# Impaired Non-Selective Response Inhibition in Obsessive-Compulsive Disorder

**DOI:** 10.3390/ijerph20021171

**Published:** 2023-01-09

**Authors:** Ruslan Masharipov, Alexander Korotkov, Irina Knyazeva, Denis Cherednichenko, Maxim Kireev

**Affiliations:** 1N.P. Bechtereva Institute of the Human Brain, Russian Academy of Sciences, Academika Pavlova Street 9, Saint Petersburg 197376, Russia; 2Institute for Cognitive Studies, Saint Petersburg State University, Saint Petersburg 197376, Russia

**Keywords:** fMRI, Go/NoGo, response inhibition, OCD, intolerance of uncertainty, psychiatry

## Abstract

Two prominent features of obsessive-compulsive disorder (OCD) are the inability to inhibit intrusive thoughts and behaviors and pathological doubt or intolerance of uncertainty. Previous study showed that uncertain context modeled by equiprobable presentation of excitatory (Go) and inhibitory (NoGo) stimuli requires non-selective response inhibition in healthy subjects. In other words, it requires transient global inhibition triggered not only by excitatory stimuli but also by inhibitory stimuli. Meanwhile, it is unknown whether OCD patients show abnormal brain activity of the non-selective response inhibition system. In order to test this assumption, we performed an fMRI study with an equiprobable Go/NoGo task involving fourteen patients with OCD and compared them with 34 healthy controls. Patients with OCD showed pathological slowness in the Go/NoGo task. The non-selective response inhibition system in OCD included all brain areas seen in healthy controls and, in addition, involved the right anterior cingulate cortex (ACC) and the anterior insula/frontal operculum (AIFO). Moreover, a between-group comparison revealed hypoactivation of brain regions within cingulo-opercular and cortico-striato-thalamo-cortical (CSTC) circuits in OCD. Among hypoactivated areas, the right ACC and the right dorsolateral prefrontal cortex (DLPFC) were associated with non-selective inhibition. Furthermore, regression analysis showed that OCD slowness was associated with decreased activation in cingulate regions and two brain areas related to non-selective inhibition: the right DLPFC and the right inferior parietal lobule (IPL). These results suggest that non-selective response inhibition is impaired in OCD, which could be a potential explanation for a relationship between inhibitory deficits and the other remarkable characteristic of OCD known as intolerance of uncertainty.

## 1. Introduction

Obsessive-compulsive disorder (OCD) is a widespread debilitating illness characterized by the occurrence of intrusive thoughts (obsessions) and repetitive behaviors or mental acts (compulsions). The lifetime prevalence of OCD is about 2.5%, which makes it the fourth most common mental illness [1,2,3]. Typically, untreated OCD is a lifelong chronic illness with recurrent peaks of symptom manifestation. Up to 40–60% of patients do not respond satisfactorily to pharmacotherapy [4]. In severe pharmacoresistant cases, neurosurgical intervention may be considered. In about 40–50% of OCD patients, cingulotomy and capsulotomy have shown a clinically prominent reduction in symptom severity [5]. In order to develop new effective methods of therapy, as well as to pinpoint the neuromarkers of this disorder [6,7], it is necessary to understand the brain mechanisms of the development and maintenance of OCD.

To date, two neurobiological hypotheses of OCD development are most common: (1) pathological hyperactivity of the error detection or action monitoring mechanism, and (2) an impairment of inhibitory control caused by imbalance in excitatory-inhibitory cortico-striato-thalamo-cortical (CSTC) pathways. According to the first hypothesis, first suggested by N.P. Bechtereva in 1971 [8,9], OCD may be caused by an impairment in the error detection mechanism. Under normal conditions, the mechanism evaluates how accurately an activity is performed, comparing the expected result to the actual result, registers disparity or an error, and sends executive signals to other brain systems in order to correct the activity and fix errors [10]. In pathology, the error detection mechanism can become an “error determinator”:
“*It can be assumed that the detector of errors acquires a sort of independent, self-sufficient significance in other, more “natural” conditions of pathology and transforms from a detector—an optimizer of activity—into the determinator of its impairments. Such a process can also underlie psychopathological syndromes, particularly those that are characterized by an obsessive repetition of actions, inappropriate behavior etc. A constant, undetermined by error, primary in relation to some action activity of the structure which usually plays the role of the detector of errors, is always signaling the inconsistency of the executed action (or any other reality) with the plan, regardless of whether the action is correct*”[8] (p. 101)

Later, a similar idea was proposed in the article “A cybernetic model of obsessive-compulsive psychopathology” by Pitman (1987), who considered the septo-hippocampal system as an “error detector” or a “comparator” [11]. In more recent works, the anterior cingulate cortex (ACC) was considered to be the key node in the error detection system that undergoes pathological morphofunctional changes in OCD [12,13,14,15,16,17,18,19,20,21,22,23].

The second hypothesis highlights the fact that patients with OCD are unable to suppress certain repetitive thoughts and actions, while they also generally demonstrate inhibition deficits [24,25,26,27,28,29,30,31,32,33,34]. It was suggested that the impairment of inhibitory control makes it difficult to suppress reactions to signals and traces of these signals that have lost their real meaning, which leads to development of stereotypical behavior. Over time, certain thoughts and actions become pathologically intrusive, that is, they develop into obsessions and compulsions. I.P. Pavlov was one of the first to propose this hypothesis in his work “An attempt at a physiological interpretation of obsessional neurosis and paranoia” (1934, [35]) that connected obsessive behavior in animals with “pathological inertness” caused by an imbalance between excitatory and inhibitory processes. In more recent studies, neuroimaging findings allowed to attribute impaired inhibitory control in OCD to imbalance between excitatory (direct) and inhibitory (indirect and hyperdirect) CSTC pathways [31,36,37,38,39,40,41].

A recent meta-analysis of functional magnetic resonance imaging (fMRI) studies by Norman et al., 2019 [34] confirmed that OCD patients show both (1) hyperactivation within the cingulo-opercular network and right prefrontal cortex during error detection and (2) hypoactivation of the cingulo-opercular and CSTC structures, including prefrontal and parietal cortices, as well as thalamic and caudate nuclei, during the performance of inhibitory control tasks (predominantly response inhibition tasks). Based on these data, a “two-component model” of OCD development can be proposed, according to which the constant false activation of the error detector leads to obsessive repetition of the same thoughts and actions that a patient is unable to suppress due to impaired inhibitory control, which leads to a “vicious cycle” in OCD.

Another prominent feature of OCD is pathological doubt, or intolerance of uncertainty [42,43,44,45,46], particularly, intolerance of ambivalent situations when two or more stimuli cause reactions of the opposite nature at the same time and to the same extent. Such a “collision” of two opposite excitatory (Go) and inhibitory (NoGo) reactions was employed in Pavlov’s laboratory to induce so called “experimental neuroses” in animals [47]. In human studies, the condition of uncertainty can be achieved in different ways, for example, using a Go/NoGo task with equally probable presentation of Go and NoGo stimuli [48,49]. In our previous study, we showed that in healthy subjects this uncertainty condition requires non-selective response inhibition, namely transient global inhibition triggered by both equiprobable NoGo and Go stimuli that prevent any premature actions [50]. In general, non-selectivity of response inhibition may be considered in two ways: as non-selectivity of inhibitory stimulus perception and non-selectivity of motor acts suppression. Firstly, inhibition may be non-selectively triggered by any imperative stimuli (both Go and NoGo stimuli), when the context is uncertain (e.g., equal probability of Go and NoGo stimuli), instead of selective triggering by NoGo stimuli (e.g., when NoGo stimuli is infrequent, as in traditional Go/NoGo paradigms) [48,49,50]. Secondly, inhibition may non-selectively affect entire motor system, instead of selectively affecting the muscles relevant to task performance [51,52,53,54]. In the current work, we consider the first type of non-selectivity.

Typically, researchers look for a selective increase in neuronal activity triggered by rare inhibitory stimuli compared to frequent Go stimuli (“NoGo > Go” effect) [49,55]. Infrequent presentation of NoGo stimuli creates a bias towards a Go response and promotes selective triggering of inhibitory process. In turn, the equiprobability of stimulus presentation leads to maximum uncertainty about the next stimulus. Several event-related fMRI studies using equiprobable Go/NoGo tasks did not reveal a statistically significant “NoGo > Go” effect associated with selective response inhibition [48,49,56,57,58]. However, the absence of statistically significant results does not mean the absence of the effect when the classical frequentist null hypothesis testing (NHST) is employed. To provide evidence for the null hypothesis (“NoGo = Go” effect), we used Bayesian parameter inference [59]. As a result, we revealed practically equivalent blood oxygenation level-dependent (BOLD) responses induced by both NoGo and Go stimuli in brain areas that, according to a meta-analysis of 20 fMRI studies, are associated with response inhibition in uncertain contexts [50].

Meanwhile, the issue of the relationship between impaired inhibitory control and intolerance of uncertainty in OCD still requires in-depth investigation [45]. Particularly, it remains unclear whether the development of this disorder is associated with an impairment of the non-selective response inhibition, which could shed light on the relationship between these two prominent features of OCD within the framework of the two-component model. Thus, the present study is aimed at testing the hypothesis of an impairment of non-selective response inhibition under conditions of contextual uncertainty in OCD patients. To do this, we utilized the equiprobable Go/NoGo task and performed the same procedures as in our previous study [50] to reveal brain areas associated with non-selective response inhibition in patients with OCD. Next, we compared OCD and healthy control groups to investigate putative impairments in non-selective response inhibition in OCD. Finally, we looked for dependencies between task performance (response times) and brain activity in OCD. We expected to register pathological slowness in inhibitory task performance (“pathological inertness”) associated with decreased activity in brain areas related to non-selective response inhibition in OCD.

## 2. Materials and Methods

### 2.1. Participants

The fMRI study included 34 healthy control (HC) participants (10 males, mean age 25.9 ± 5.2) and 14 patients with OCD (nine males, mean age 30.0 ± 10.6). The difference in mean age between groups was not significant (Student’s *t* = 1.8, *p* = 0.0776). The proportion of females to males was higher in the HC group (χ^2^ = 5.04, *p* = 0.0247). All participants were right-handed [60]. HC and OCD participants were excluded if they had: (1) a history of brain trauma or neurological disease, (2) alcohol or substance abuse, (3) contraindications to MRI scanning. HC participants were excluded if they reported any history of mental illness. All OCD patients were diagnosed according to ICD-10 (International Classification of Diseases). Twelve OCD patients were undergoing pharmacotherapy at the time of testing, including one or more of the following: antidepressants of the selective serotonin reuptake inhibitor class (escitalopram), tricyclic antidepressants (clomipramine), benzodiazepine anxiolytics (phenazepam), typical antipsychotics (zuclopenthixol), anticonvulsants (carbamazepine, valproate) and atypical antipsychotics (aripiprazole). Two patients were drug-free at the time of testing. The average Yale–Brown obsessive-compulsive scale (Y-BOCS) score for the OCD group was 18.7 ± 7.2 (obsessive subscale 9.7 ± 5.6, compulsive subscale 9.0 ± 5.8), which corresponds to moderate symptoms [61]. The average duration of the OCD was 9.3 ± 6.8 years. All participants signed an informed written consent and were paid for their participation. The study procedures were performed in accordance with the Declaration of Helsinki, and the study protocol was approved by the Ethics Committee of N.P. Bechtereva Institute of the Human Brain, Russian Academy of Sciences (ethical protocols dated 18 April 2013 and 30 December 2014).

### 2.2. Task Design

The cued equiprobable Go/NoGo task (see Figure 1) was described in detail in Masharipov et al., 2022 [50]. Each task trial consisted of cue and target stimuli presented for 100 ms. The cue-target interval was 1000 ms. The intertrial interval (ITI) varied from 2800 to 3200 ms with an increment step of 100 ms. The stimuli were images of animals and plants presented in different combinations corresponding to Go, NoGo and Ignore trials.

The fMRI study consisted of two task sessions with distinct instructions. According to the first instruction (see Figure 1A), the participants should press the response button with the right thumb as soon as possible upon a presentation of the image pair “animal-animal” (“A-A Go” trials) and refrain from acting upon a presentation of the pair “animal-plant” (“A-P NoGo” trials). According to the second instruction (see Figure 1B), the participants act after a presentation of the pair “animal-plant” (“A-P Go” trials) and suppress an action upon a presentation of the pair “animal-animal” (“A-A NoGo” trials). In both experiments, if the first presented stimulus was an image of a “plant,” the subjects did not need to take any actions in response to a presentation of any second stimuli of a trial (“P-A Ignore” and “P-P Ignore” trials). During two sessions, the participants were randomly presented with 100 NoGo trials, 100 Go trials, 100 P-A Ignore trials, and 100 P-P Ignore trials. In the absence of stimuli, a fixation cross was displayed in the centre of the screen. Additionally, to improve design efficiency, 100 zero events (fixation crosses) were randomly inserted between trials. Zero event duration varied from 3000 to 5000 ms with an increment size of 500 ms.

The subjects performed a training task just before scanning to ensure they understood the instructions. Additionally, before each fMRI session, the subjects were reminded of the need to react to Go stimuli as quickly as possible while refraining from reacting to NoGo stimuli. The Invivo’s Eloquence fMRI System (Invivo, Orlando, FL, USA) was used to present stimuli, record responses, and synchronize with fMRI data acquisition. The task presentation was programmed in E-prime 2.0 software package (Psychology Software Tools Inc., Pittsburgh, PA, USA).

### 2.3. Data Acquisition

Neuroimaging data were acquired on a Philips Achieva 3.0 Tesla scanner (Philips Medical Systems, Best, The Netherlands). The structural T1-images were registered using a T1-weighted 3D fast field echo (T1W-3D-FFE) sequence with the following parameters: repetition time (TR)—25 ms, echo time (TE)—2.2 ms, flip angle—30°, field of view (FOV)—240 × 240 mm, 130 axial slices, voxel size—1 × 1 × 1 mm. The functional T2*-images were obtained using a single-pulse echo planar imaging (EPI) sequence with the following parameters: TR = 2 s, TE = 35 ms, flip angle—90°, FOV—200 × 186 mm, 31 axial slices, voxel size—3 × 3 × 3 mm. Two dummy scans were performed prior to each session. To minimize head movements, we used an MR-compatible soft cervical collar and foam padding.

### 2.4. Data Preprocessing

Functional T2*-images were realigned to the first image of the session, slice time corrected, coregistered to the anatomical image, segmented, normalized to an MNI template, and smoothed with 8 mm full-width half-maximum (FWHM) isotropic Gaussian kernel. Preprocessing and data analysis were performed using the SPM12 software (http://www.fil.ion.ucl.ac.uk/spm, accessed on 7 January 2023).

### 2.5. Data Analysis

The selective response inhibition hypothesis predicts an increase in the neuronal activity in NoGo trials compared to Go trials (“NoGo > Go”). At the same time, the non-selective response inhibition hypothesis suggests a practically equivalent increase in the neuronal activity in response to presentation of both NoGo and Go stimuli (“NoGo = Go”) in the brain areas related to response inhibition when the context is uncertain. To reveal brain areas with practically equivalent neuronal activity in NoGo and Go trials (“NoGo = Go”), one has to provide evidence for the null hypothesis.

Classical frequentist null hypothesis significance testing (NHST) considers the probability of obtaining the observed data, or more extreme data (*D*+), given that the null hypothesis (*H*_0_) is true. If this probability is lower than the alpha level *P*(*D*+|*H*_0_) < α, then we can ‘reject the null hypothesis. That is, NHST based on probabilistic ‘proof by contradiction’ and can only provide some evidence against the null hypothesis but cannot provide evidence for the null hypothesis. To avoid this problem, we can calculate the probability that the null hypothesis is true given the obtained data (*D*). If this ‘inverse probability’ is above a predefined probability threshold *P*(*H*_0_|*D*) > *P_thr_* (usually *P_thr_* = 95%), then we can “accept the null hypothesis”. This probability is also known as the posterior probability and can be calculated for both null and alternative hypothesis using Bayes’ rule: *P*(*H*|*D*) = (*P*(*D*|*H*)·*P*(*H*))/*P*(*D*), where *P*(*D*|*H*) is the probability of obtaining the exact data given the hypothesis or the likelihood (do not confuse with *P*(*D*+|*H*)), *P*(*H*) is the prior probability of the hypothesis (our knowledge of the hypothesis before we obtain the data), and *P*(*D*) is a normalizing constant ensuring that the sum of posterior probabilities over all possible hypotheses equals one (marginal likelihood). In other words, according to Bayes’ rule the posterior probability is proportional to the product of likelihood and prior probability. This means that with Bayes’ rule, we update our prior beliefs about the hypothesis based on the obtained data. Bayesian approaches directly provide evidence for the null and alternative hypotheses given the data [59,62].

In order to independently localize brain areas associated with response inhibition in uncertain context, we exploited the results of our recent meta-analysis of fMRI studies that used equal probability Go/NoGo tasks [50]. The meta-analysis included activation foci for “50/50% Go/NoGo blocks > 100% Go-control blocks” contrasts from twenty fMRI studies. It was performed with the activation likelihood estimation (ALE) approach [63] and validated using the Seed-based *d* Mapping with Permutation of Subject Images (SDM-PSI) approach [64]. All corresponding details of the meta-analysis are described in our previous work [50], and its results are publicly available online on NeuroVault (https://neurovault.org/collections/6009, accessed on 7 January 2023).

In order to identify brain areas associated with non-selective response inhibition control, we looked for overlap between the results of the meta-analysis of equiprobable Go/NoGo studies (“50/50% Go/NoGo blocks > 100% Go-control blocks”) and the results of Bayesian analysis of acquired fMRI data for OCD group (“NoGo = Go”).

It is important to note that the “50/50% Go/NoGo blocks > 100% Go-control blocks” contrast reveals areas not only related to response inhibition, but also non-inhibitory cognitive functions, such as preparatory attentional processes and uncertainty perception. We used the cue-target Go/NoGo paradigm, which was designed to separate preparatory attentional processes from the execution and suppression of prepared motor programs [7]. The first (cue) stimuli trigger attentional preparatory activity, and the second (target) stimuli trigger prepotent motor acts and inhibitory processes that suppress them [7,48,49,56]. In the current work, we considered hemodynamic responses evoked by the target stimuli. To do this, we used onset times of second stimuli presentation to create regressors of the general linear model (GLM) for each subject at the first level of analysis. Additionally, we added six head motion regressors in the GLM to account for the movement artefacts [65]. The “NoGo-Go” contrasts from OCD patients were used as variables to test hypotheses on selective and non-selective response inhibition at the second level of analysis. We used a gray matter binary mask based on the segmentation of each subject’s structural T1-images, as we did not expect to detect inhibitory-related signal changes in white matter. For the “NoGo-Go” comparison, we used both classical NHST and Bayesian parameter inference (BPI) implemented in the BayInf toolbox based on SPM12 (https://github.com/Masharipov/Bayesian_inference, accessed on 7 January 2023) [59]. For the second-level Bayesian analysis, SPM12 implements the hierarchical parametric empirical Bayes approach with the global shrinkage prior [66]. It represents a prior belief that, on average, in the whole brain, there is no global experimental effect (BOLD signal change for the “NoGo-Go” contrast). This is based on the fact that any change in neuronal activity evoked by task stimuli occurs locally in a limited set of voxels. If the posterior probability of finding the effect exceeding the effect size threshold, *θ* > γ, is greater than the predefined probability threshold, *P_thr_* = 95%, then the hypothesis on the presence of “NoGo > Go” effect will be accepted for a particular voxel. If the effect falls within the region of practical equivalence (ROPE), *θ* ∈ [−γ; γ], with a probability of *P_thr_* = 0.95, then the hypothesis of the null “NoGo = Go” effect will be accepted. The hypothesis on the presence of the “Go > NoGo” effect will be accepted if the posterior probability of finding the effect that does not exceed the negative effect size threshold, *θ* < −γ, is greater than *P_thr_* = 95%. If none of the above criteria are satisfied, the data in particular voxel are insufficient to distinguish the null hypothesis from the alternative hypothesis (“low-confidence” voxels). The group-level effect size threshold γ was set at one standard deviation of the prior variance of the contrast, which is the default in SPM12 [66]. A simulation study showed that this threshold provides high sensitivity to both “activated” and “not activated” voxels while protecting against incorrect decisions [59]. The one prior SD threshold typically detect similar “activations” as classical NHST inference with a voxel-wise family-wise error (FWE) corrected threshold of 0.05 [59]. For visualization purposes, the posterior probabilities were converted to the logarithmic posterior odds (LPO). LPO of 3 correspond to a posterior probability of 0.95.

To evaluate non-selective response inhibition impairments in OCD group compared to HC group, we used “Go + NoGo” contrasts, as non-selective inhibition should appear in both Go and NoGo trials. The between-group comparison was performed using classical frequentist NHST with a primary significance threshold of 0.005 and cluster-level FWE-corrected threshold of 0.05. We used the more common frequentist inference for the between-group comparison because we did not aim to look for practically equivalent neuronal activity in both groups. Finally, to look for dependencies between response times (RTs) and brain activity in NoGo and Go trials (“Go + NoGo” contrasts), we used a linear regression model with one regressor corresponding to a linear contrast of interest (“Go + NoGo”) and another regressor corresponding to individual mean RTs centered to the overall mean. For this analysis, we also used a primary significance threshold of 0.005 and cluster-level FWE-corrected threshold of 0.05. Anatomical localization of clusters was identified using the xjView toolbox (http://www.alivelearn.net/xjview, accessed on 7 January 2023).

## 3. Results

### 3.1. Behavioural Data

The mean response omission in the Go trials was 2.97 ± 3.92% for the HC group and 3.21 ± 3.45% for the OCD group. The mean of false alarms in NoGo trials was 0.35 ± 0.64% for the HC group and 0.43 ± 0.94% for the OCD group. The between-group differences in response omission and false alarms were not statistically significant. The mean response time (RT) was 384 ± 60 ms for the HC group and 491.1 ± 117.5 ms for the OCD group. The between-group difference in RT was statistically significant (Student’s *t* = 4.17, *p* < 0.001, Cohen’s d = 1.33).

### 3.2. FMRI Data

Classical NHST was unable to detect a statistically significant increase in NoGo trials compared to Go trials for the OCD group both with an FWE-corrected threshold of 0.05 and with a more liberal uncorrected threshold of 0.005. Bayesian analysis did not reveal selective response inhibition in “NoGo > Go” contrast as well. The reversed “Go > NoGo” contrast revealed left-hemisphere dominant motor activations (see Figure 2A, red color). Additionally, Bayesian analysis allowed us to determine brain areas with practically equivalent activity in equiprobable NoGo and Go trials (see Figure 2A, green color). The null “NoGo = Go” effect was revealed for a widely distributed set of regions throughout the entire brain surrounding “Go > NoGo” clusters and separated from them by “low-confidence” voxels (see Figure 2A, white color).

The meta-analysis of 20 fMRI studies using equiprobable Go/NoGo tasks published in Masharipov et al., 2022 [50] revealed several brain areas associated with response inhibition in uncertain context (see Figure 2B): right dorsolateral prefrontal cortex (DLPFC), right inferior parietal lobule (IPL), right temporoparietal junction (TPJ), bilateral inferior frontal gyrus (IFG) and anterior insula (also known as anterior insula/frontal operculum (AIFO)), right premotor cortex (PMC) and frontal eye field (FEF), bilateral anterior cingulate cortex (ACC) and supplementary motor area (SMA), and bilateral thalamus. In the current study, we found an overlap between the results of the meta-analysis and brain areas with practically equivalent activity in NoGo and Go trials (“NoGo = Go”) identified in the OCD group by Bayesian analysis (see Figure 2C and Table 1).

Thus, the following brain areas correspond to the nodes of the non-selective response inhibition system in patients with OCD: the right ACC, DLPFC, IPL, TPJ, PMC, FEF, and bilateral AIFO. Note, that the same brain areas, except for the right ACC and right AIFO, were found in the HC group in the previous study [50].

The direct between-group comparison revealed hypoactivation of several dorsal CSTC structures in OCD: bilateral DLPFC, SMA, ACC, as well as thalamic and caudate nuclei (see Figure 3A and Table 2). Importantly, there was an overlap between hypoactive brain areas and non-selective response inhibition areas in OCD within the right ACC (Brodmann area 32) and right DLPFC (Brodmann area 9) (see Figure 3B).

The linear regression revealed an inverse dependency between brain activity in equiprobable NoGo and Go trials (“Go + NoGo” contrast) and patients’ mean RT in several brain regions: bilateral ACC, middle cingulate cortex, posterior cingulate cortex, precuneus, right IPL, and right DLPFC (see Figure 4A and Table 3). The overlap between these areas and the non-selective response inhibition areas in OCD was found within the right IPL (Brodmann area 40) and the right DLPFC (Brodmann area 9) (see Figure 4B).

## 4. Discussion

The results obtained in the present study speak in favor of the hypothesis about the impairment of non-selective response inhibition in conditions of uncertain context. It was shown that the mean RT was increased in the OCD patients compared to the HC subjects while performing a response inhibition task in the condition of equiprobable presentation of Go and NoGo stimuli. The response slowness in OCD patients may be referred to as the “pathological inertness” described by I.P. Pavlov in the “experimental neurosis” model caused by the “collision” of excitatory and inhibitory stimuli [35,47]. The increase in RT in various inhibitory control tasks has been previously reported in the literature (see a meta-analysis by Norman et al., 2019 [34]). A psychological study by Aycicegi et al., 2003 [24] with an equiprobable Go/NoGo design also found a significant RT increase in OCD compared to HC. Another psychological study by Soref et al., 2008 [26] that used an equiprobable Flanker task showed a significant increase in response time in HC subjects with a high tendency to OCD symptoms compared to a group with a low tendency. Moreover, meta-analyses of behavioral studies show a general decrease in the speed of motor reactions [30] and a decrease in the speed of information processing [28] in patients with OCD. In addition, some studies associate higher severity of Not Just Right Experiences (NJREs) with slower RT in OCD patients performing Go/NoGo tasks [27].

The current research is the first work that discovers the brain areas related to non-selective response inhibition in OCD patients. This set of brain areas was almost identical to the previously revealed one for the HC control group and included: the right DLPFC, IPL, TPJ, PMC, FEF, and the left AIFO [50]. However, two additional nodes of the non-selective response inhibition system have been found in OCD patients: the right ACC and the right AIFO. In addition, a direct between-group comparison showed hypoactivation in OCD patients within several cingulo-opercular and dorsal CSTC structures: bilateral ACC, SMA, DLPFC, thalamus, and caudate nuclei. This result is consistent with a meta-analysis of fMRI studies using various inhibitory control tasks [34] that also showed hypoactivity of the cingulo-opercular and CSTC regions in OCD. A recent multimodal meta-analysis also found that cingulo-opercular and striatal regions are hyperactivated across different major psychiatric disorders (including OCD) in distinct inhibitory control tasks [67]. At the same time, the revealed abnormal brain activity in OCD during inhibitory task performance complements the results from resting-state functional connectivity (rsFC) studies, which revealed dysfunctional connectivity within the CSTC and cingulo-opercular circuits in OCD [31]. These studies revealed both decreased and increased rsFC in OCD patients depending on the seed region and OCD subpopulation (for review see [68]). For instance, an rs-fMRI study by Jang et al., 2010 [69] showed hypoconnectivity between seed in PCC and ACC, DLPFC and basal ganglia compared to the HC group. At the same time, a study by Harrison et al., 2009 [70] found decreased rsFC between the dorsal striatum and DLPFC but increased rsFC along the ventral corticostriatal axis. An rs-fMRI study on OCD development by Fitzgerald et al., 2011 [71], which included child, adolescent, and adult subpopulations, revealed that OCD in the youngest patients was associated with reduced connectivity of the striatum and thalamus to ACC. Another rs-fMRI study by Vaghi et al., 2016 [72] showed that reduced rsFC between the caudate and ventrolateral PFC (VLPFC) was associated with reduced cognitive flexibility, while reduced FC between the putamen and DLPFC was associated with symptom severity and decreased goal-directed task performance.

An important addition to the previous studies is that we found an overlap between hypoactivated structures and brain areas related to non-selective response inhibition in patients with OCD. These were the right ACC, which was not observed in the HC group [50], and the right DLPFC previously associated with the non-selective response inhibition network in healthy subjects. Remarkably, the ACC also is the key node of the error detection system that exhibits hyperactivity during error processing, according to fMRI meta-analysis [34], and glucose hypermetabolism in the resting state, according to our previous positron emission tomography study [73].

Finally, we revealed a significant dependency between OCD slowness and hypoactivation during inhibitory control. The higher mean RT was associated with lower BOLD signal in equiprobable Go and NoGo trials within bilateral ACC, middle cingulate cortex, posterior cingulate cortex, precuneus, right IPL, and right DLPFC. Moreover, an overlap was found between these areas and non-selective response inhibition areas in the right DLPFC and right IPL. Since the subjects were instructed to respond to Go stimuli as quickly as possible, the RT was taken as a measure of the inhibitory task performance effectiveness [74,75]. When the context is uncertain, the need for response inhibition arises both during suppression (NoGo trials) and execution of prepared actions (Go trials); therefore, the decreased neuronal activity in the non-selective response inhibition nodes (right IPL and right DLPFC) may be associated with the less efficient operation of response inhibition and pathological slowness. The decrease in the computational effectiveness of non-selective response inhibition can be explained by the fact that it takes more time to «release» the transient global inhibitory process and open access to the prepared motor program.

Combined with previous neuroimaging studies, the results of our research allow us to link the impairment of inhibitory control with another prominent characteristic of OCD, intolerance of uncertainty, within the framework of the proposed “two-component” model of OCD development. It can be assumed that in OCD, hyperactivity of the error detection system, including ACC, may be referred to as undetermined or misspecified error signals. These constant error signals induce anxiety and a subjective sensation that “something is wrong” or NJREs [8,11,76]. NJREs are increased in the absence of a sufficient amount of information, and, as it is known, a prominent feature of OCD is intolerance of uncertainty. In order to decrease anxiety and uncertainty, the patient often performs futile actions which only lead to temporary relief. Temporary relief reinforces such behavior, which becomes stereotypical and ritualized as a result. However, ultimately these actions do not correct the false error. The constant false activation of the detector of errors leads to obsessive repetition of the same actions and thoughts, which the patient is unable to suppress due to insufficient inhibitory control. The non-selective response inhibition does not work efficiently enough and takes more time. Since it also works in Go trials, it takes more time (compared to the healthy subjects) to “release” inhibition for the subsequent implementation of the action.

## 5. Conclusions

In the current study, we have for the first time demonstrated a dysfunction of the non-selective response inhibition system in OCD patients that operates under conditions of context uncertainty in the cued Go/NoGo task. The uncertain context was created by the “collision” of two equiprobable excitatory (Go) and inhibitory (NoGo) stimuli. Under such conditions, the inhibitory process is non-selectively triggered by both types of stimuli. The altered non-selective inhibitory control in patients with OCD was associated with the extension of the non-selective response inhibition network. Compared to healthy controls, the non-selective response inhibition in OCD involved two additional regions in the right ACC and the right AIFO. At the same time, a direct comparison between the OCD and the HC groups revealed hypoactivity in OCD within two regions related to non-selective response inhibition: the right ACC and the right DLPFC. The decrease in the efficiency of the inhibitory task performance manifested as a reaction speed slowdown in the OCD patients compared to healthy controls. Regression analysis allowed us to connect pathological slowness to decreased brain activity in the cingulate regions, as well as the right DLPFC and the right IPL, related to non-selective response inhibition. The current findings extend previously observed hypoactivity of the cingulo-operular and CSTC regions in OCD during selective response inhibition and cognitive inhibition. Impairment of non-selective response inhibition sheds light on the link between inhibitory deficits and intolerance of uncertainty seen in OCD.

## 6. Limitations and Further Considerations

The first limitation of the current study is a relatively small sample size for the OCD group. However, the obtained sample size was enough to reveal statistically significant differences between OCD and HC groups after the correction for multiple comparisons. The larger samples would potentially allow to reveal OCD subgroups, and their relevance to non-selective response inhibition impairment could be explored.

The second limitation is the question of how precisely observed impairments are related to inhibitory processes per se. In the current study, we used cue-target design to separate preparatory attentional processes from the executive motor and inhibitory processes. However, the processes of non-selective inhibition, uncertainty perception and attentional orienting may be near impossible to disentangle, because orienting to salient task-related signals is inevitably accompanied by broad motor suppression [54]. Both processes are fast and transient since subthreshold prepotent responses are inhibited as early as around 150 ms after target stimuli presentation [48,54,77,78]. Furthermore, attentional orienting and inhibitory processes may rely on overlapping neural circuits. A possible solution would be to identify brain activity causally related to inhibitory effects at the level of the primary motor cortex, spinal cord, and effector muscles (e.g., using dynamic causal modeling and/or transcranial magnetic stimulation).

Finally, future research should consider how the identified hypoactivation, which has been related to impaired non-selective response inhibition and uncertainty perception, is associated with aberrant functional and effective connectivity between cortical and subcortical structures that form inhibitory and excitatory CSTC circuits.

## Figures and Tables

**Figure 1 ijerph-20-01171-f001:**
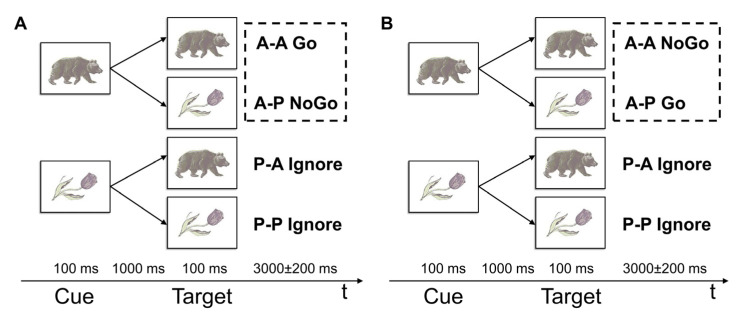
Task design. S1—the first stimulus (preparatory cue). S2—the second stimulus (target). A—images of animals, P—images of plants. (**A**) The first instruction: “A-A Go”, “A-P NoGo”. (**B**) The second instruction: “A-P Go”, “A-A NoGo”. Dashed boxes highlight trials that were compared to test the hypothesis on non-selective response inhibition.

**Figure 2 ijerph-20-01171-f002:**
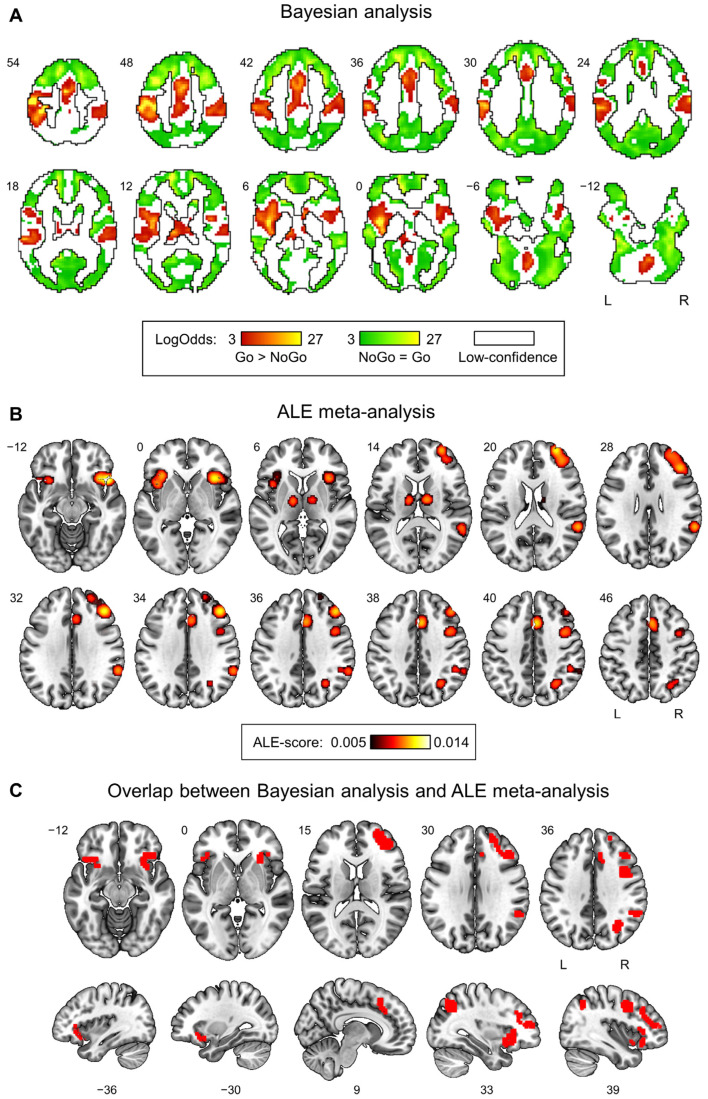
Non-selective response inhibition in the OCD group. (**A**) Results of Bayesian analysis of fMRI data from the patients with OCD. The “Go > NoGo” effect is shown in red. The “NoGo > Go” effect was not detected. The null “NoGo = Go” effect is indicated in green. The “low confidence” voxels are shown in white. (**B**) The result of the meta-analysis of 20 fMRI studies using equal probability Go/NoGo tasks (“50/50% Go/NoGo blocks > 100% Go-control blocks” contrast) from Masharipov et al., 2022 [50]. (**C**) Overlap between the results of Bayesian analysis (“NoGo = Go”) and the ALE meta-analysis.

**Figure 3 ijerph-20-01171-f003:**
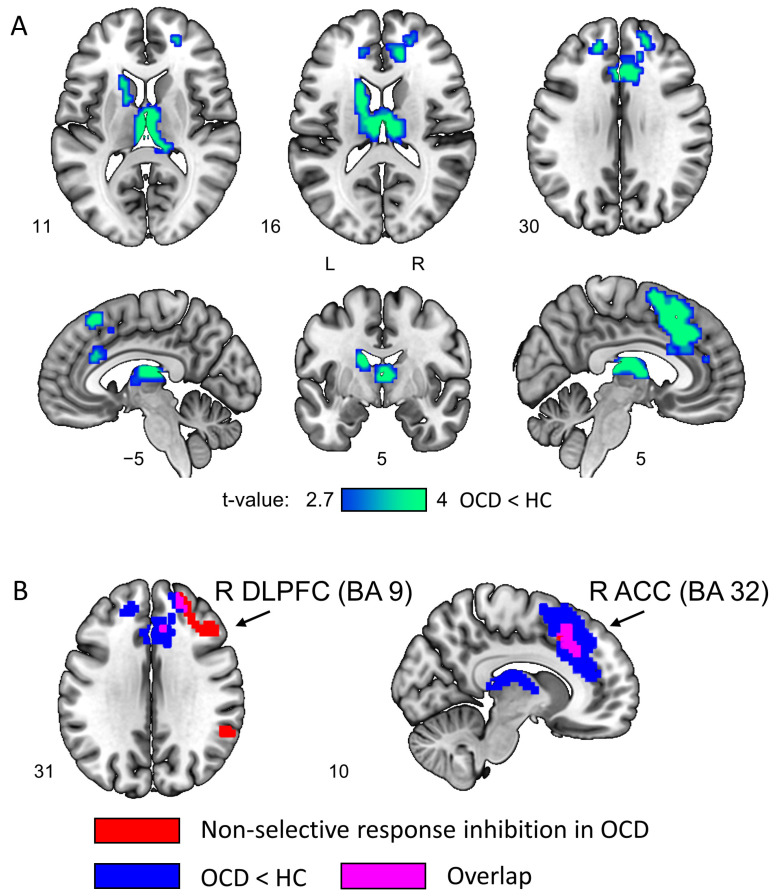
(**A**) Hypoactive brain areas during equiprobable Go/NoGo task performance in the OCD group compared to the HC group (“Go + NoGo” contrast). Cluster-level FWE-corrected threshold of 0.05. (**B**) Nodes of the non-selective response inhibition system in patients with OCD (red color), hypoactivated brain areas in OCD (blue color), and the overlap between them (violet color).

**Figure 4 ijerph-20-01171-f004:**
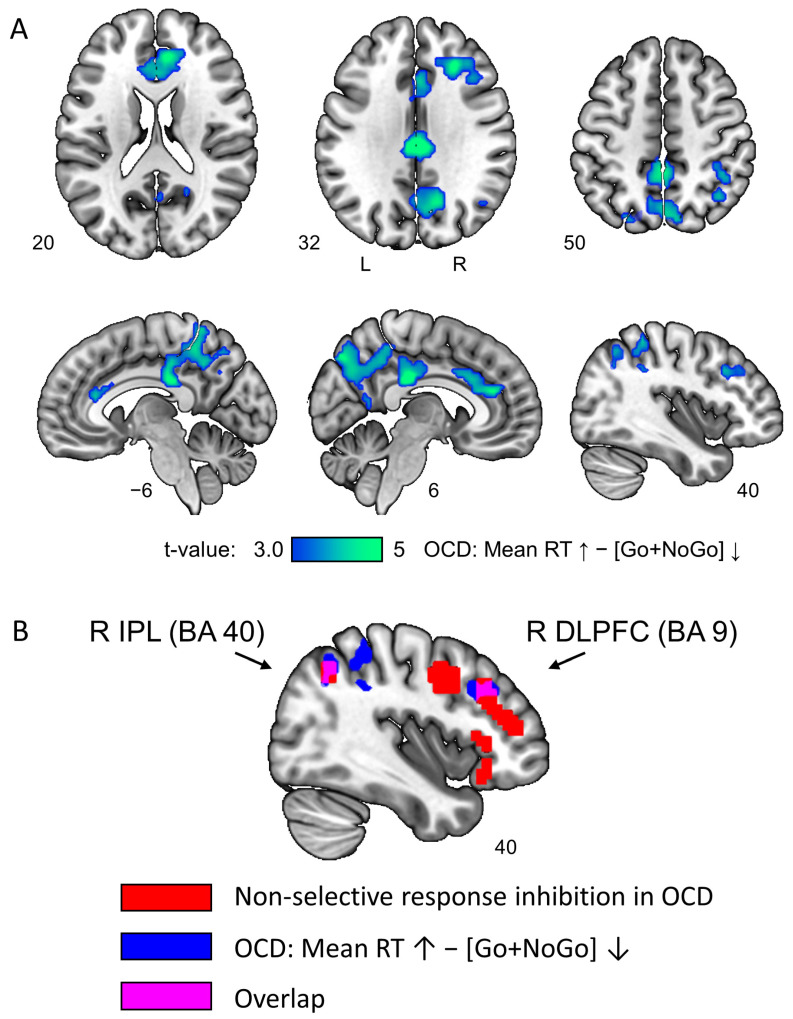
(**A**) Inverse dependency between patients’ mean RT and brain activity in equiprobable Go and NoGo trials (“Go + NoGo” contrast). Cluster-level FWE-corrected threshold of 0.05. (**B**) Nodes of the non-selective response inhibition system in patients with OCD (red color), brain areas associated with response slowness in OCD (blue color), and the overlap between them (violet color).

**Table 1 ijerph-20-01171-t001:** Nodes of the non-selective response inhibition system in patients with OCD. Overlap of the results from a meta-analysis of 20 fMRI studies (“50/50% Go/NoGo > 100% Go control”) and Bayesian analysis of obtained fMRI data from OCD patients (“NoGo = Go”).

No.	Cluster Size, mm^3^	Anatomical Localization *	Centroid Coordinates (MNI), mm		
			x	y	z
1	3591	R: DLPFC, BA 9, 10	36	42	24
2	1971	R: Premotor cortex, FEF, BA 6, 8	39	9	42
3	1647	R: IFG, Anterior insula, BA 13, 47	30	24	0
4	1539	R: IPL, BA 7, 40	30	−60	45
5	891	L: IFG, Anterior insula, BA 13, 47	−36	24	−9
6	621	R: TPJ, BA 40	54	−45	36
7	405	R: ACC, BA 32	6	27	45

* Abbreviations: ACC—anterior cingulate cortex; DLPFC, dorsolateral prefrontal cortex; IFG, inferior frontal gyrus; IPL—inferior parietal lobule; FEF—frontal eye field; TPJ—temporoparietal junction; R/L—right/left hemisphere; BA—Brodmann area.

**Table 2 ijerph-20-01171-t002:** Decreased BOLD-signal in equiprobable Go and NoGo trials (“Go + NoGo” contrast) in the OCD group compared to the HC group. Cluster-level FWE-corrected threshold of 0.05.

No.	Cluster Size, mm^3^	Anatomical Localization *	T-Value	Coordinates of Local Maxima (MNI), mm		
				x	y	z
1	1614	R: ACC, BA 32	4.53	6	26	35
		R: Superior frontal gyrus, SMA, BA 6, 8	4.34	12	20	56
		R: SMA, BA 6	4.24	6	8	62
		L: Superior frontal gyrus, SMA, BA 6, 8	3.94	−6	29	56
		R/L: ACC, SMA, BA 6, 32	3.66	3	14	47
		L: Superior frontal gyrus, BA 9	3.49	−18	44	23
		R: Superior frontal gyrus, BA 9	2.97	18	50	29
2	7614	L/R: Thalamus	5.01	0	−16	14
		L: Caudate nucleus	4.89	−15	−1	20

* Abbreviations: ACC—anterior cingulate cortex; SMA—supplementary motor area; TPJ—temporoparietal junction; R/L—right/left hemisphere; BA—Brodmann area.

**Table 3 ijerph-20-01171-t003:** Significant dependency between increased patients’ mean RT and decreased BOLD-signal in equiprobable Go and NoGo trials (“Go + NoGo” contrast). Cluster-level FWE-corrected threshold of 0.05.

No.	Cluster Size, mm^3^	Anatomical Localization *	T-Value	Coordinates of Local Maxima (MNI), mm		
				x	y	z
1	29,322	R: MCC, BA 23	9.21	6	−25	29
		L: MCC, BA 23	7.73	−3	−25	29
		L: Precuneus, SPL, BA 7	6.03	−15	−73	44
		R: Precuneus, SPL, BA 7	5.51	6	−76	47
		R/L: Paracentral lobule, BA 5	5.22	0	−43	53
		R: Precuneus, BA 7	4.98	15	−64	32
		R: SPL, BA 7	4.69	30	−55	62
		R: IPL, BA 40	4.51	39	−40	56
		R/L: PCC, BA 23	3.58	3	−55	11
		R: IPL, BA 40	3.13	42	−58	50
		R: DLPFC, BA 9	7.79	30	35	35
2	9288	R: ACC, BA 32	5.65	9	38	20
		R: ACC, BA 24	4.40	3	20	29
		L: ACC, BA 24	4.39	−6	29	17

* Abbreviations: ACC—anterior cingulate cortex; MCC—middle cingulate cortex; PCC—posterior cingulate cortex; DLPFC—dorsolateral prefrontal cortex; SPL—superior parietal lobule; IPL—inferior parietal lobule; R/L—right/left hemisphere; BA—Brodmann area.

## Data Availability

The data analyzed in the current study are available from the corresponding author on reasonable request.

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
