# Peer review of "Impaired Non-Selective Response Inhibition in Obsessive-Compulsive Disorder"

_ijerph, 2023, doi:10.3390/ijerph20021171_

Round 1

Reviewer 1 Report

Thank you for the opportunity to review this manuscript. The authors explore the brain areas associated with ‘non-selective response inhibition’ in individuals with OCD vs. healthy controls. Overall, I think that this is an interesting topic, but I have some major concerns regarding the study design and interpretation of results. I organize the comments below by section.

Introduction

I would explicitly spell out what you mean by 'non-selective response inhibition'. I take you to mean a domain-general response inhibition system that is activated during maximum uncertainty (e.g. equiprobably presentation of Go/NoGo trials) and not just traditional Go/NoGo paradigms that create a bias towards a Go response and promote selective triggering of the inhibitory process.

What is your hypothesis? Are you going to test the two-component model of OCD and if so, how?

Methods:

You say the groups are matched on age and gender, but there is clearly a majority of female participants in the HC group and a minority of female participants in the OCD group. Please clarify.

What pharmacotherapies were the OCD patients taking?

Please describe the Bayesian analyses (2nd to last paragraph) in more layman's terms for the reader that is not acquainted with this approach.

In the methods, you write "In order to identify brain areas associated with non-selective response inhibition control, we looked for overlap between the results of the meta-analysis of equiprobable Go/NoGo studies (“50/50% Go/NoGo blocks > 100% Go-control blocks”) and the results of Bayesian analysis of acquired fMRI data for OCD group (“NoGo = Go”)...To evaluate non-selective response inhibition impairments in OCD group compared to HC group, we used “Go + NoGo” contrasts, as non-selective inhibition should appear in both Go and NoGo trials." I don't get the rationale. Won't NoGo trials still recruit more neural structures of motor response inhibition than NoGo trials? Wouldn't the "50/50% Go/NoGo blocks > 100% Go-control blocks” design just give an idea of the brain processes underlying uncertainty and not "non-selective response inhibition”?

Results

What was the rationale for the linear regression in Fig4, and what are the details of how it was conducted?

Discussion

I just have the general comment I am not sure that the results point to the "brain areas related to non-selective response inhibition in OCD patients." Based on the statistical design, it may reveal more about the brain areas associated with "anticipation of making or inhibiting a motor response in uncertain contexts."

Reviewer 2 Report

In this timely report, the authors used a Go/NoGo task to investigate response inhibition in OCD. They demonstrated impaired DLPFC/ACC and frontostriatal functions that contribute to non-selective response inhibition difficulties in OCD.

Overall, the study is interesting and methodologically sound. I have a few questions and remarks.

1. Sample size: It is important because psychiatric neuroimaging suffers from inadequate statistical power. The authors should emphasize this issue in a limitation section.

2. Did the authors plan to analyze the data for functional and effective connectivity?

3. How can be the results interpreted in the framework of resting-state fMRI results from other studies?

4. How about the effects of medications? 
